# Current Cardioprotective Strategies for the Prevention of Radiation-Induced Cardiotoxicity in Left-Sided Breast Cancer Patients

**DOI:** 10.3390/jpm13071038

**Published:** 2023-06-24

**Authors:** Vasiliki Nikovia, Evangelos Chinis, Areti Gkantaifi, Maria Marketou, Michalis Mazonakis, Nikolaos Charalampakis, Dimitrios Mavroudis, Kornilia Vasiliki Orfanidou, Antonios Varveris, Chrysostomos Antoniadis, Maria Tolia

**Affiliations:** 1Medical School, University of Crete, Vassilika, 71110 Heraklion, Greece; 2Radiotherapy Department, Theagenio Anticancer Hospital of Thessaloniki, 54639 Thessaloniki, Greece; 3Cardiology Department, University General Hospital of Heraklion, Heraklion, 71110 Heraklion, Greece; 4Department of Medical Physics, Faculty of Medicine, University of Cret, Iraklion, P.O. Box 2208, 71003 Heraklion, Greece; 5Oncology Clinic, Metaxa Cancer Hospital, Mpotasi 51, 18537 Piraeus, Greece; 6Department of Medical Oncology, University General Hospital of Heraklion, 71500 Heraklion, Greece; 7Department of Radiotherapy, University Hospital/Medical School, University of Crete, Vassilika, 71110 Heraklion, Greecemariatolia@uoc.gr (M.T.)

**Keywords:** radiation-induced cardiotoxicity, newer radiotherapy methods, cardioprotection

## Abstract

Background: Breast cancer (BC) is the most common malignancy in females, accounting for the majority of cancer-related deaths worldwide. There is well-established understanding about the effective role of radiotherapy (RT) in BC therapeutic strategies, offering a better local–regional control, prolonged survival, and improved quality of life for patients. However, it has been proven that conventional RT modalities, especially in left-sided BC cases, are unable to avoid the administration of high RT doses to the heart, thus resulting in cardiotoxicity and promoting long-term cardiovascular diseases (CVD). Recent radiotherapeutic techniques, characterized by dosimetric dose restrictions, target volume revision/modifications, an increased awareness of risk factors, and consistent follow-ups, have created an advantageous context for a significant decrease inpost-RT CVD incidence. Aim: This review presents the fundamental role of current cardioprotective strategies in the prevention of cardiotoxic effects in left-BCRT. Material and Methods: A literature search was conducted up to January 2023 using the Cochrane Central Register of Controlled Trials and PubMed Central databases. Our review refers to new radiotherapeutic techniques carried out on patients after BC surgery. Specifically, a dose evaluation of the heart and left anterior descending coronary artery (LADCA) was pointed out for all the included studies, depending on the implemented RT modality, bed positioning, and internal mammary lymph nodes radiation. Results: Several studies reporting improved heart sparing with new RT techniques in BC patients were searched. In addition to the RT modality, which definitely determines the feasibility of achieving lower doses for the organs at risk (OARs), better target coverage, dose conformity and homogeneity, and the patient’s position, characteristics, and anatomy may also affect the evaluated RT dose to the whole heart and its substructures. Conclusions: Modern BC RT techniques seem to enable the administration of lower doses to the OARs without compromising on the target coverage. The analysis of several anatomical parameters and the assessment of cardiac biomarkers potentiate the protective effect of these new irradiation modalities, providing a holistic approach to the radiation-associated risks of cardiac disease for BC patients. Despite technological advances, an inevitable cardiac radiation risk still exists, while adverse cardiac events may be observed even many years after RT. Studies with longer follow-ups are required in order to determine the effectiveness of modern breast RT techniques.

## 1. Introduction

Breast cancer (BC) constitutes the leading female malignancy (2.3 million diagnosed cases every year) and most common cancer-related death in women worldwide [1]. In total, 19.6 million disability-adjusted life years (DALYs) out of 107.8 million DALYs deriving from female malignant neoplasms are globally attributed to BC [2].Improved survival and clinical outcomes have been demonstrated in recent decades [3], owing to more advanced screening methods/earlier diagnoses, improved available tools, and the comprehensible awareness-raising of related risk factors.

The majority of BCs are unilateral [4], with a slightly higher prevalence of left-sided BC but no significant evidence on the specific causes of BC laterality [5].However, several studied factors could be responsible for this observed laterality, such as breast-feeding patterns, a pattern of cerebral asymmetry, handedness, breast size and density, the impact of hormonal fluctuation on the mammary glands, genetic predisposition, and sleeping habits, etc. [6]. A well-emphasized factor involved inunilateral cancer cases is alack of stability during breast development or alaterality, namely a fluctuating asymmetry between the left and right breast [4].

RT is indicated for the majority of BC-conserving surgery cases, enhancing long-term survival rates [7,8,9,10]. Left-sided BC patients, who receive neoadjuvant [11] or adjuvant RT, may present a higher risk of CVD development [12,13,14].Left-sided BCs present metastases to the regional lymph nodes less commonly than right-sided cancers [5] and require the delivery of a more extended RT field. Left-sided BC is related to worse clinical outcomes and higher death rates than right-sided BC [15]. RT may induce damage to the endothelial cells’ microvascular environment, activating the inflammatory cascade of atherothrombosis, with a subsequent occlusion of blood flow, ischemia or infarction, myocardial cell death, and fibrosis [16]. At the same time, it provokes inflammation, oxidative stress, the thickening and fibrosis of vessels media adventitia, and consequential atherosclerosis in the large vessels such as the coronary and carotid arteries.

The main clinical manifestations of RT-induced cardiotoxicity include pericarditis, myocarditis, arrhythmia, coronary artery disease, myocardial infarction, congestive heart failure, valvular heart disease, and cardiomyopathy [11]. The risk of cardiotoxicity and CVD-induced mortality may be even higher when specific risk factors coexist. Emphasis should be given to the treatment of cardiovascular risk factors that increase cardiac side effects, such as diabetes mellitus, arterial hypertension, dyslipidemia, obesity, age, smoking, a sedentary lifestyle, and unhealthy nutrition, and to the control of underlying cardiovascular diseases [17].

Obesity, dyslipidemia, excessive alcohol consumption, tobacco use, hormone replacement therapy, insufficient physical exercise, the consumption of processed food, and psychological stress are classified as modifiable risk factors, and age (over 40), gender (female), race, genetic profile, a family history of BC, previous breast diseases, age at menarche, number of pregnancies and births, breast density, and previous radiation exposure as non-modifiable factors [13,18,19,20,21]. However, even with anelimination of all the modifiable factors, the maximum risk decrease achieved would be approximately 30% [2].

The great number of cardiac death sattributed to RT’s late cardiotoxic effects, observed after a long-term follow-up, necessitates the development of optimal RT methods, providing safety and a better quality of life for patients [22,23]. The concept of personalized medicine refers to the tailoring of irradiation to each left-sided BC patient’s unique characteristics. Therefore, the aim of the present review is to highlight the availability of current cardioprotective strategies concerning their potential heart-sparing benefit.

## 2. Materials and Methods

We included all papers offering any data concerning new radiotherapeutic techniques with dosimetric dose restrictions to the heart and LADCA carried out on patients after BC surgery. Electronic databases were searched using synonyms for “left breast cancer”, “radiotherapy”, “irradiation”, “cardiotoxicity”, and “radiation-induced cardiotoxicity” in women with BC. A filter was not used, since any type of study was considered, without a restriction to randomized controlled trials. PubMed and the Cochrane Database of Controlled Trials were searched up until January 2023 with a restriction to English-written publications. The articles were procured for a meticulous inspection and evaluation of their contiguity. Cross references from the included studies were hand searched. All the titles and abstracts retrieved were printed and reviewed manually and independently by two reviewers. The studies that clearly did not meet the inclusion criteria were excluded. Finally, original articles, literature reviews, trials, and meta-analyses were included in our review.

## 3. Results

We identified the studies reporting on the availability of current cardioprotective strategies that could be applied to newly diagnosed left-sided BC patients for the prevention of radiation-induced cardiotoxicity. We searched only in PubMED/Central. In addition to the optimal RT method, which definitely determines the feasibility of achieving lower doses for the organs at risk (OARs), the patient’s position, parameters, and anatomy may also affect the evaluated RT dose to the whole heart and its substructures. Detailed data of the included studies are summarized in Table 1 and Table 2.

## 4. Discussion

Although a great number of studies have extensively analyzed late cardiotoxicity after left-breast RT, the data about early subclinical cardiac changes remain scarce [32]. Characteristically, in 2019, Gkantaifi et al. clearly illustrated how important the introduction of new tools is for the detection of early cardiotoxicity in left-BC patients after RT [33]. There is a definite need to focus on the roles of strain echocardiographic imaging and specific cardiac biomarkers concerning their potential to predictearly cardiac dysfunction, thus leading to an earlier detection of patients at risk, better clinical outcomes, and an improved quality of life. To be more specific, strain echocardiographic imaging has been found to be a sensitive tool for predicting early cardiac damage after chemotherapy, as well as in various conditions [34,35,36,37]. However, its role in detecting early cardiotoxicity after breast RT is still questionable, with reported decreased levels of strain parameters from a few weeks to 2 years after RT [38,39]. Thus, these changes might predict early cardiac abnormalities, demonstrating the usefulness of strain rate imaging. Furthermore, measuring specific biomarkers in BC patients’ serum during or after RT could contribute to the prediction, diagnosis, and risk assessment of cardiovascular disease [40].Studies have shown that the appropriate biomarkers in patients undergoing RT include cardiac troponin T (cTnT), cardiac troponin I, and natriuretic peptides such as B-type natriuretic peptide (BNP) and N-terminal pro B-type natriuretic peptide (NT-proBNP) (cTnI). Elevated levels of these molecules may indicate increased myocardial stretch and thus may be used as biomarkers for left ventricular dysfunction and heart failure [41,42]. Although such biomarkers are insufficient on their own to determine the path of treatment for each patient, they should be used as a trigger or follow-up tool for the attending physician [43].

## 5. Relevant Sections

### 5.1. Natural Antioxidants

Natural products have also attracted much scientific interest for their potential impact on the prevention of radiation side effects, thus reducing the risk of heart tissue damage. Indeed, the pathophysiology of oxidative stress and the production of free radicals, such as reactive oxygen species (ROS) and reactive nitrogen species (RNS), possess major roles in RT-induced cardiovascular disease [44,45]. Free radicals react with cell macromolecules, leading to DNA and protein oxidation and lipid peroxidation, provoking irreversible damage to cells and disrupting normal cells or tissue function and homeostasis. Thus, the development and use of agents that could eliminate these free radicals or limit their activity would prevent and repair the irradiation side effects, reducing the risk of heart tissue injury. Studies have illustrated some natural antioxidants that may inhibit the free radicals’ damage and reduce the incidence of heart disease, such as:Vitamin E: reduces nicotinamide adenine dinucleotide phosphate (NADPH) oxidase activity, inhibits lipid peroxidation, and downregulates the nuclear factor kappa-light-chain-enhancer of activated B cells (NF-κB).Silymarin: stimulates nuclear factor erythroid 2–related factor 2(Nrf2) expression and downregulates NF-κB.Resveratrol: reduces NADPH oxidase uncoupling and endothelial nitric oxide (eNOS) and upregulates antioxidant defense enzymes.Lycopene: scavenges singlet oxygen, sulfur, nitrogen dioxide, and sulfonyl-free radicals.Melatonin: stimulates the enzymatic antioxidant system.Hesperidin: passivates NADPH oxidase and inhibits Transforming growth factor beta 1 (TGF-β1) mRNA expression.Curcumin: (a) attenuates the IL-4 protein upregulation and its receptor, IL4Ra1, and (b) the expression of the dual oxidases (Duox1 and Duox2) responsible for extracellular matrix stabilization via oxidative cross-linking.Zingerone: (a) decreases malondialdehyde levels, (b) increases glutathione/catalase activity, (c) reduces inflammatory markers, i.e., tumor necrosis factor-alpha, cardiac myeloperoxidase activity, and cyclooxygenase-2 protein, and (d) reduces caspase-3 gene expression and nuclear DNA fragmentation.Caffeic acid phenethyl ester (CAPE): reduces the activities of (a) malondialdehyde (MDA), (b) xanthine oxidase (XO), and (c) adenosine deaminase (ADA), and increases the levels of (d) nitrate/nitrite (NO(x) and (e) superoxide dismutase SOD in heart tissue [45,46].

### 5.2. Pharmacological Drugs

Furthermore, the development of pharmacological drugs represents another emerging option for the prevention of late cardiotoxicity. Even though limited research has proven their benefit, studies have revealed that the administration of metformin can be cardioprotective in left-sided BC patients who undergo adjuvant RT [46,47,48,49]. Characteristically, metformin has a great number of actions related to endothelium function, such as antifibrotic effects, impeding the formation of the inflammatory environment and oxidative stress in endothelial cells, the promotion of endothelial cell proliferation and differentiation, vascular tone regulation, and the arrest of endothelial to mesenchymal transition. Studies have suggested the administration of metformin for left-sided BC and declared that biguanide not only prevents RICT toxicity, but the probability of coronary artery disease can also be significantly reduced [25,50,51,52,53]. Yu JM et al. [49], in a retrospective national cohort study, prescribed metformin for >28 days (defined daily dose) to2.062 early-stage left-sided BC patients. The authors found that metformin use during adjuvant breast irradiation significantly reduced the risk of major heart events (adjusted hazard ratio [aHR], 0.789; 95% confidence interval [CI], 0.645–0.965; *p* = 0.021).

### 5.3. Modern Radiotherapy Techniques

Before the widespread use of three-dimensional conformal RT (3DCRT), heart irradiation was inevitable [54,55]. Characteristically, the anterior, anteroseptal, and anterolateral left ventricular walls, and even more, their apical parts and the LADCA, were exposed to higher radiation doses in left-sided BC cases [26,56,57]. RT for left BC may involve some incidental heart exposure to ionizing radiation. Modern RT methods can drastically reduce the effect of this exposure on the subsequent risk of heart disease [58]. In order to minimize adverse cardiac toxicity, a remarkable effort has been made to develop new RT techniques, so as to achieve reduced heart irradiation without simultaneously compromising the target coverage.

Research has revealed an association between left ventricular function and RT dose and a linear relationship between coronary artery disease and radiation dose to cardiac tissue, as well as an association between myocardial infarction or ischemic heart disease and mean heart dose (MHD) [59]. Darby et al. demonstrated that the CVD risk increased linearly with the MHD. The risk was 7.4% per gray, with no clear threshold below which there was no risk. The risk started within the first 5 years after irradiation and continued for at least 20 years. Ref. [18] while a few years later, Skytta and colleagues reported increased serum Troponin T levels with reported MHD 4Gy [60].Additionally, in a recent small prospective study, Gkantaifi et al. highlighted the need for assessment in RT treatment planning, additionally to whole heart volume, as well as dose constraints for cardiac substructures—such as the left anterior descending artery (LAD) and left ventricle (LV)—for patient’s improved heart protection [61]. Actually, although the radiation exposure to the whole heart and left ventricle was low in the whole study, there was a reported higher exposure to the LAD, raising major concerns about the vulnerability of the latter structure.

However, there has not yet been determined a threshold above which a significant risk of cardiac toxicity may occur; therefore, an increased need for continual dose fluctuation is still observed. In this context, the majority of new RT methods and revised treatment planning protocols focus on are duction in their radiation of OARs and consequently in radiation-induced cardiotoxicity (RICT) [62]. Except for classic free breath 3D conformal RT, the new radiotherapeutic methods that will be further developed below include prone positioning, respiratory-gating and breath-hold RT, deep-inspiration breath hold (DIBH) and gating, forward-planned and inverse-planned intensity modulated RT (IMRT), volumetric-modulated arc therapy (VMAT), helical tomotherapy, and proton therapy [53].

#### 5.3.1. Prone Positioning

Research supports that a prone set-up may reduce heart irradiation, as it allows for the breast to fall from the chest wall while it is restrained by the bed, increasing, in this way, the distance between the heart and irradiated breast tissue [63]. The ability of a meticulously planned prone RT is of vital significance, so as to minimize the treatment fields without penetrating the heart orjeopardizing the target coverage. Previous studies have reported that the benefit of prone positioning during BC RT was controversial due to the anterior translocation of the heart, indicating that only the ipsilateral lung may benefit from it [64,65,66]. Other studies have demonstrated that mainly women with large-sized breasts would profit from this method, not only because their breast would be compressed in the supine set-up and not homogenously irradiated, but compared to small-sized breast patients, the distance between the OARS and breast would be greater. Characteristically, in 2014, Varga Z et al. [65] included 83 left BC patients in their study comparing RT plans in both prone and supine positions, aiming to indicate the optimal tool for the ideal treatment position for BC patients. Actually, many patients presented a high exposure to the heart and LAD in the prone position, illustrating the significant role of patient’s anatomy, BMI, and breast size [34]. Nevertheless, recent studies have suggested that prone RT predicts less long-term cardiotoxicity, independently of a patient’s breast size, in comparisonto the supine set-up, with both the heart and ipsilateral lung receiving lower doses [67,68].

#### 5.3.2. Respiratory Gating Techniques

Respiratory gating (RG) techniques, including deep inspiration breath hold (DIBH), have been widely used during the last few years for postoperative left-sided BC RT, as more studies have increasingly proven reduced heart irradiation and a consequently reduced risk for RT-induced cardiac disease with these methods [56,62]. The two most commonly used DIBH techniques include voluntary DIBH (vDIBH) and involuntary or moderate DIBH (iDIBH). iDIBH requires active breathing control (ABC) devices, which control airflow and halt it over specific volumes [69]. In contrast, vDIBH is based on patient guidance, in order to monitor their respiratory cycle. During the treatment and delivery of radiation, patients are trained to inhale deep and hold their breath for a specific period of time. The aim of administering radiation during a deep breath is to ensure that it occurs when the distance between the breast and chest wall is at a maximum level during the breath cycle, as the heart moves posteriorly and inferiorly, being influenced by diaphragm flattening and lung expansion [70,71]. DIBH can reduce high-dose areas and the mean heart dose by 33–66% from the initial RT dose, also providing an improvement in LADCA and lung dose. In a recent study by Tang L et al., including 11 left BC patients undergoing CT either in free breathing or DIBH, a significantly reduced dose to the heart and LADCA was found with DIBH. Furthermore, compared to 3DCRT plans, VMAT achieved a significant dose reduction [27]. In 2021, Ferdinard et al. conducted a prospective studywith31 left BC patients, indicating a significantly reduced dose to the heart and LAD with the DIBH technique compared to free breathing (MHD 2.4 Gy vs. 4.01 Gy) [24]. Moreover, Sakyanun P et al. and Ferini G et al. illustrated a decreased radiation exposure to the heart and LAD by applying the DIBH technique compared to free breathing in left BC patients. The maximum heart distance and breast size were found to influence the MHD [16,72]. Apart from the foregoing, Darby et al. indicated that the incidence of major coronary events is increased by 7.4%for each 1Gy in MHD, with no determined threshold [18]. On the contrary, Taylor et al. showed a 4% increase in CVD mortality per Gy MHD, defining the dose of 4 Gy as the threshold, below which there is no extra risk of CVD mortality [73]. Although DIBH is the most frequently applied cardio sparing RT technique, many physicians and institutions mention that an increased treatment time is required, consequently inducing an additional workload [71]. As a result, meticulous patient selection is necessary and should be based on maximum heart distance, parasagittal cardiac contact distance, tumor characteristics, a patient’s anatomical features, the technique convenience, patient compliance, accurate training, and tolerance [74]. Currently, a great number of studies have concluded that DIBH is superior to Free Breathing RT regarding simultaneous cardioprotection and target coverage [59,62].Thus, many researchers have suggested that the delivery of the DIBH technique constitutes a more effective heart-sparing method for left-sided BC cases [24,74,75]. In a recent retrospective study from Ngujen et al., a reduced heart dose with the DIBH technique was demonstrated, even in cases of internal mammary nodes irradiation (−56.4%) [31]. During DIBH, both in simulation and an RT course, the patient takes a deep breath for a period of time. An appropriate inspiration and chest excursion allow for lung expansion and diaphragm flattening. DIBH takes the advantage of a more favorable heart position anatomy, displacing the heart from the tangential radiation fields.

Regarding the method of breath adapting RT (BART), enhanced inspiration gating (EIG) is also included. During EIG, patients are audio-coached throughout the entire training session (including CT scanning, set-up imaging, and RT course). The patient breathes deeper than normal, but unlike for DIBH, the patient does not perform normal breathing between the deep breaths. A real-time positioning management system (marker block with six reflective markers placed on the chest of the patient, and infrared light reflected by these markers that is detected by a camera to monitor the antero-posterior chest movement) is used to monitor the patients’ breathing. The patients are instructed to take a deep breath, adjusting separately the duration of inspiration and expiration. This RT technique administers lower radiation doses to the OARs in comparison to the Free Breathing technique, reducing the risk of long-term cardiovascular effects. Although some studies have proven that both RG techniques offer similar cardioprotective action and do not compromise the target coverage, others have indicated that DIBH achieves greater reductions in the MHD and LADCA dose [33,62]. Actually, the retrospective study from Edvardsson et al., regarding 32 patients undergoing tangential or locoregional adjuvant breast free-breathing RT, reported a decreased median V25Gy for the heart and LAD by applying enhanced inspiration gating, from 2.2% to 0.2% and 40.2% to 0.1% (*p* < 0.001), respectively [25].

#### 5.3.3. Proton Beam Therapy

Proton Beam Therapy (PBT) has been used in clinical medicine since the mid 1980s, but not to a large extent; therefore, it poses limitations such as the cost of treatment facilities, as well as dose and range uncertainties due to radiation’s sensitivity to the density of the substances it crosses [76,77]. Proton RT is a rapidly evolving irradiation modality because of its energy deposition according to Bragg Peaks, which means reduced entrance and exit doses and the administration of peak energy to the target volume, in an overall more advantageous dose-depth distribution [78]. As a consequence, PBT can be selectively applied to patients with left-sided BC in order to avoid heart exposure to irradiation. In 2018, the study results from Luo L et al. about postmastectomy RT with protons reported promising conclusions on local control and toxicity in a 3-year follow-up [30].

A major PBT constriction is the high cost and big size of the gantry, impeding its widespread use and availability. Fixed proton beamlines (FBLs) aim to overcome this obstacle by permitting the implementation of proton RT in conventional treatment rooms, which is not an easily available approach. There is limited evidence proving PBT’s advantages over other heart-sparing radiation techniques, so further research is required for proton RT application in patients achieving maximum benefits.

#### 5.3.4. Intensity Modulated Radiotherapy

Intensity Modulated RT (IMRT) represents an advanced form of 3D-Conformal RT that alters the intensity of multiple photon beams and delivers them from diverse directions, creating highly conformal dose distributions [33,79]. With regard to the literature, IMRT achieves the administration of lower doses to a greater proportion of cardiac tissue, thus resulting in an improved sparing of the OARs, dose uniformity, and the coverage of the target volume in comparison to3D-CRT [80].Studies have illustrated that the irradiation of the internal mammary chain (IMC) in left-sided BC patients increases the cardiac dose [81].

Treatment with IMRT is particularly beneficial for node-positive BC patients, as it intensifies the positive impact of RT on the overall survival in cases with positive internal mammary lymph nodes. There are two types of IMRT: forward-planned IMRT (FP-IMRT) and inverse-planned IMRT (IP-IMRT). Both of them may achieve a similar plan quality to IMRT and are considered as more suitable for lymph node irradiation [69].

#### 5.3.5. Volumetric Modulated Arc Therapy

Volumetric modulated arc therapy (VMAT) represents another modern RT technique that is optimal for left BC cases. VMAT is an IP-IMRT technique that combines the variation in gantry rotation speed, the on-treatment movement of MLC leaves, and the delivery of radiation in less time by rotating the machine around the patient. Studies have shown that VMAT is beneficial for left-sided BC patients by administering lower doses to healthy tissues, including the heart and coronary arteries, and avoiding the induced cardiac dysfunction [36]. In 2021,Zhang et al. enrolled 30 patients undergoing RT after a mastectomy [81].Compared to IMRT, VMAT treatment plans resulted in better target volume coverage and reduced heart tissue exposure to radiation.

#### 5.3.6. Helical Tomotherapy

Another technique that belongs to the category of arc-based methods is called Tomotherapy. Tomotherapy can be further divided into axial, serial, or helical tomotherapy (HT) [82]. Studies have shown that HT reduces the MHD and the dose to LADCA by delivering RT in a continuous spiral axis [83]. However, HT is mainly recommended for patients whose treatment plan does not achieve the desirable dose limits by applying other sparing techniques to the heart tissue [84]. Indeed, in a study from Nichols GP et al., regarding 15 patients undergoing postmastectomy RT, improved OARS sparing was shown at low doses (<5Gy) with the VMAT technique compared to HT, while the latter was more effective at higher doses [37].

#### 5.3.7. Surface-Guided Radiation Therapy

Left BC patient setup with Surface-Guided Radiation Therapy (SGRT) may improve the isocenter reproducibility in comparison to three or four tattoos [85]. The main SGRT advantage is that it utilizes thousands of points on a patient’s body and does not prolong the treatment time or clinical workflow [86]. Therefore, more accurate patient positioning and intrafractional monitoring is possible with the use of SGRT rather than lasers and the heart position deviations are smaller [85,86,87,88].

## 6. Conclusions

In conclusion, newer RT techniques are currently being used for preventing the development of further cardiovascular diseases in BC patients. The majority of studies have concluded that the choice of optimal treatment technique and modality depends on which of them are available in each department, as well as on individualized patient data concerning anatomy, dosimetry, and tumor characteristics. Undoubtedly, the identification of selection criteria and prognostic factors is imperative, aiming at the optimal short-term and long-term benefits for each patient.

## 7. Future Directions

There has clearlybeen observed an increased scientific interest in the improvement of modern available radioprotective techniques for reduced dose administration to the heart tissue and coronary vessels. However, the most effective method has yet to be determined. Despite the long latency period until the clinical interpretation of these results, a reduction in CVD morbidity and mortality is expected. Current research supports that the optimal treatment plan is based on a combination of factors, such as patient-related parameters (tumor characteristics, breast structure, a patient’s physical state, the existence of risk factors, etc.), the availability of each RT technique, and the possibility for curative-intent treatment. Consequently, several clinical questions have arisen about the use of heart-sparing techniques in everyday clinical practice. Waiting for the medical community to provide more precise answers, further well-organized studies, meta-analyses, and clinical trials have to be performed.

## Figures and Tables

**Table 1 jpm-13-01038-t001:** Dosimetry of the dose distribution for breast cancer radiotherapy techniques, considering the parameters of the mean dose and maximum dose regarding the heart and LADCA.

First Author	Number of Treated Patients	Target	Delivery Technique		D_mean/heart (Gy)_		D_max/heart_ _(Gy)_	D_mean/LADCA_ _(Gy)_	D_max/LADCA_ _(Gy)_	*p* _value_
Ferdinand S et al. [24]	31 (Comparison of both techniques in the same patients, before treatment)	Breast or CW in left-sided BC patients after BCS and in those with high RFs after MRM	FB/DIBH		4		39.4	12.6	31.9	39.15%reduction in mean heart dose in DIBH compared to FB(2.4 Gy vs. 4.01Gy) *p* < 0.001
					2.4		31.5	8.7	25.8	19% reduction in maximum LAD dose and a 9.9% reduction in ipsilateral lung mean dose *p* = 0.036
Edvardsson A et al. [25]	32 (Comparison of both techniques in the same patients, before treatment)	16 patients: tangential breast irradiation to the WB after lumpectomy, 9 patients: locoregional RT after lumpectomy, 7 patients: locoregional RT after mastectomy	FB/EIG		3.1		-	25.4	-	Median V25Gy for heart, LAD decreased for EIG (2.2 to 0.2% and 40.2 to 0.1%, respectively) (*p* < 0.001)
					1.5		-	8.0	-	MedianV25heart and LAD decreased for EIG (3.3 to 0.2% and 51.4 to 5.1%, respectively(*p* < 0.001)
Varga Z et al. [26]	83 (Comparison of both positions in the same patients, before treatment)	3D-Conformal RT in left-sided BC patients.	Prone/Supine		Prone 2.18 ± 0.15 /Supine 2.89 ± 0.19		-	Prone 11.06 ± 0.79/ Supine 13.7 ± 0.79	-	Mean Heart Dose: Prone 2.18 ± 0.15 Supine 2.89 ± 0.19 <0.001 LAD Mean dose: 11.06 ± 0.79/ Supine 13.7 ± 0.79 *p* value = 0.014
Tang L et al. [27]	11	Left-sided BC patients after BCS	DIBH-3DCRT/FB-3DCRT		0.88/1.52		13.80/29.33	2.21/3.04	5.94/12.29	DIBH-3DCRT, dose reductions in all evaluation parameters of the heart and LADCA compared with those in the FB (*p* < 0.05).
	All patients were compared in FB and DIBH positions and in 3DCRT and VMAT techniques	Whole Breast								
			DB-VMAT		0.47		5.32	1.43	3.76	DIBH–VMAT dose reductions in heart and LADCA compared to 3DCRT plans (*p* < 0.05).
Zhang Y et al. [28]	30 (Both IMRT and VMAT plans were created for each patient)	CW and IMN in left-sided BC patients after radical mastectomy	IMRT/ VMAT		12.6 ± 0.7/11.5 ± 0.7		-	-	-	≤0.002
Nichols GP [29]	15 (Both HT and VMAT plans were created for each patient)	CW and RLN after mastectomy and previously treated with HT (7 patients: bilateral CW and 8 patients: left CW)	HT/VMAT		14.4 ± 0.6/12.9 ± 0.5		-	-	42.2 ± 1.6/38.3 ± 1.3	<0.05
Luo L. et al. [30]	36 All patients received proton therapy	Postmastectomy WB in left-sided BC	3DCPT FB/DIBH		0.84 Gy (RBE) (range 0–3.2 Gy)		3.2 Gy (RBE)	-	-	-
Nguyen MH et al. [31]	49 (All patients were compared with both FB and DIBH treatment plans)	Left-sided BC patients after mastectomy or BCS 49 of whom received RNI	FB/DIBH	IMN treated		6.73	-	-	-	DIBH reduced the average mean heart dose from 6.73 Gy to 2.79 Gy in the IMN treated group and from 4.77 Gy to 1.55 Gy in the IMN untreated group compared to FB technique (both *p* < 0.001).
				IMN untreated		4.77	-	-	-	
				IMN treated		2.79	-	-	-	
				IMN untreated		1.55	-	-	-	
Sakyanun P et al. [16]	25 (All plans were created with FB and DIBH techniques and four radiation treatment plans)	WB, WB, and IMN in left-sided BC patients after BCS	FB/ DIBH	IMN treated		10.24	-	31.98	-	<0.0001
				IMN untreated		5.38	-	19.84	-	<0.0001
				IMN treated		6.43	-	23.88	-	<0.0001
				IMN untreated		2.95	-	11.48	-	<0.0001

Abbreviations:CW, chest wall; BC, breast cancer; RFs, risk factors; BCS, breast conservation surgery; MRM, modified radical mastectomy; FB, free breathing; DIBH, deep-inspiration breath hold; EIG, enhanced inspiration gating; WB, whole breast; RT, radiotherapy; 3DCRT, three-dimensional conformal radiotherapy; DB, deep breathing; VMAT, volumetric modulated arc therapy; IMN, internal mammary nodes; IMRT, intensity modulated RT; RLN, regional lymph nodes; HT, helical tomotherapy; 3DCPT, three-dimensional conformal proton therapy; RNI, regional node irradiation; and LADCA: left anterior descending coronary artery. Dmean/heart (whole heart mean dose in Gy), Dmax/heart (whole heart max dose in Gy), Dmean/LADCA (left anterior descending coronary arterymean dose in Gy), and Dmax/LAD dose(left anterior descending coronary arterymax dose in Gy).

**Table 2 jpm-13-01038-t002:** Dosimetry of the dose distribution for breast cancer radiotherapy techniques, considering the parameters of V_5_ (%), V_10_ (%), V_20_ (%), V_25_ (%), and V_40_ (%) regarding the heart, and V_20_ (%) and V_25_ (%) regarding the LADCA.

First Author	Delivery Technique		Heart V_5_ (%)	Heart V_10_ (%)	Heart V_20_ (%)	Heart V_25_ (%)	Heart V_40_ (%)	LAD V_20_ (%)	LAD V_25_ (%)
Ferdinand S et al. [24]	FB		14.2	8.9	-	-	-	-	-
	DIBH		7.6	3.4	-	-	-	-	-
	*p* _value_		0.00	0.00	-	-	-	-	-
Edvardsson A et al. [25]	FB		-	-	-	3.3	-	-	51.4
	EIG		-	-	-	0.2	-	-	5.1
	*p* _value_		-	-	-	<0.001	-	-	<0.001
Varga Z et al. [26]	Prone		-	-	-	2.01	-	21.91	-
	Supine		-	-	-	3.54	-	29.26	-
	*p* _value_		-	-	-	0.010	-	<0.001	-
Tang L et al. [27]	FB-3DCRT		2.68	-	-	-	-	-	-
	DB-3DCRT		0.23	-	-	-	-	-	-
	DB-VMAT		0.04	-	-	-	-	-	-
	*p* _value_		<0.01	-	-	-	-	-	-
Zhang Y et al. [28]	IMRT/VMAT		27.9/20.9	12.2/9.1	7.4/4.6	-	2.0/0.7	-	-
	*p* _value_		<0.001	<0.001	<0.001	-	<0.001	-	-
Luo L et al. [30]	3DCPT		4.3	-	0.5	-	-	-	-
Nguyen MH et al. [31]	FB	IMN treated	34.5	19.9	8.9	6.4	-	-	-
		IMN untreated	19.4	11.6	6.3	5.0	-	-	-
	DIBH	IMN treated	12.7	4.1	1.0	0.4	-	-	-
		IMN untreated	2.8	0.8	0.2	0.1	-	-	-
	*p* _value_		<0.001	<0.001	<0.001	<0.001	-	-	-
Sakyanun P et al. [16]	FB	IMN treated	-	-	-	16.96	-	13.96	-
		IMN untreated	-	-	-	8.20	-	6.22	-
	DIBH	IMN treated	-	-	-	10.37	-	7.50	-
		IMN untreated	-	-	-	3.48	-	2.48	-
	*p* _value_		-	-	-	<0.0001	-	<0.0001	-

Abbreviations: V_5_ (%), the percentage volume of organ receiving 5 Gy; V_10_ (%), the percentage volume of organ receiving 10 Gy; V_20_ (%), the percentage volume of organ receiving 20 Gy; V_25_ (%), the percentage volume of organ receiving 25 Gy; V_40_ (%), the percentage volume of organ receiving 40 Gy.

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
