# Peer review of "Current Cardioprotective Strategies for the Prevention of Radiation-Induced Cardiotoxicity in Left-Sided Breast Cancer Patients"

_jpm, 2023, doi:10.3390/jpm13071038_

Round 1

Reviewer 1 Report

Dear authors,

I found your work quite interesting, but it needs some improvements:

1) In Line 86, you mentioned neoadjuvant radiotherapy, which is not currently the standard of care for breast cancer, although some encouraging findings already exist (PMID: 36011025). If this mention was intentionally made, please, add a pertinent reference as that one previously suggested.

2) In the section about "natural oxidants", you made an introduction longer than the content actually interesting for practitioners. Please, expand more and more the lines from 165 to 168 with practical suggestions and a discussion of evidence from the studies you cited.

3) Line 173: When is metformin cardioprotective? Before, during, or after RT administration? Please, provide more useful indications on such evidence. Does an approved posology exist for the study scenario?

4) Line 185: the RT role in prolonging overall survival in BC patients is well established. Therefore, the assumption that "the toxicity of RT might overweigh its benefit" is definitively not true. Moreover, the cited reference is not pertinent to support this thesis. Please, rephrase.

5) Line 189: infraction must be corrected in infarction. Infraction was erroneously used somewhere else. Please, re-check the entire manuscript.

6) Line 191: Please, clarify whether the 2 Gy MHD is the safety threshold suggested by Darby. As I remember, even for only 1 Gy MHD there is still a risk of 7.4% CVD.

7) In line 201, where "Except for 3D conformal RT" is, please, specify classic free breath 3D conformal RT. Also, the DIBH technique may be planned using the 3D- CRT.

8) In line 219, the term "less long-term cardiotoxicity" suggests somewhat availability of clinical results, whereas this is not true since both cited studies [60, 61] are only dosimetric studies. Maybe, "predicts" instead of "induces" is more correct. Please, rephrase.

9) In line 237, you erroneously indicated the name "Pitchaya" instead of the surname "Sakyanun". Please, correct this. The findings of these authors, especially the breast size, agree with those of Ferini et al. Please, cite and discuss PMID: 33788746.

10) In line 246, after "... features, technique convenience and tolerance.", please, cite PMID: 35884538 as pertinent literature. You discussed the same issues in that paper.

11) In line 249, cite PMID: 34988301 together with reference 66.

12) From line 250 to line 255, clarify which differences are between DIBH and EIG techniques.

13) Values in line 257 are confusing. Rephrase the whole sentence. In line 258, do you mean "settings" (tangential or locoregional) instead of "techniques"? I didn't understand.

14) Lines from 282 to 284 are totally wrong since Nguyen used 3D-CRT, not IMRT! This citation and sentence are out of context. Delete and possibly move where appropriate.

15) Lines 291-292 "and requires IMC irradiation along with IMRT" does not make sense at all. Rephrase.

16) Line 311, the acronym "SBRT" is wrong. You mean "SGRT". Please, correct it.

17) Line 320, it would be preferable to not cite references in the conclusion section. Delete them.

18) Please, re-check the correctness of the tables.

Reviewer 2 Report

The manuscript entitled Newer cardioprotective personalized strategies for the prevention of radiation-induced cardiotoxicity in left-sided breast cancer patientsreviewed the literature with aims to summarize the role of modern methods on the prevention of cardiotoxic effects of left breast cancer patients.

General Critique:

From a general standpoint, the current review is lack of the clear research point. Which is the major concern of the manuscript? Prevention of CV risks from radiation from the advancement of radiotherapy machine in modern era, personalized immobilization during raditherapy (DIBH, EIG, etc), antit-oxidants, or pharmacological drugs? However, the manuscript also have not performed standard review criteria, including basic search strategy, time period of literature search, search strategies, and how to assess or validate the quality of each published trial report or cases series? So, in current form of manuscript is like a summarized report of journal reading, not a review article.

The basic question is interesting. However, I would like to raise the following points:

1.     “Newer cardioprotective personalized strategies…” Is there any new finding in current review about the topic of “Newer or personalized?” There is still lack of “cardioprotective personalized strategies” in the article. The topic might be more appropriate.to bed revised:  “The current cardioprotective strategies…”

2.     The section of Introduction is too simplified, not well-organized, and not focus study aims. The author should be stated more clearly, thoroughly, and be focus on the points of factors on the dosimetry from radiation exposure, specific technique for breath-hold…  The order of cited references should be cautious and re-organized.  

3.     Please add the flow chart of search strategy according PRISMA as a basis for reporting systematic reviews. Is there any definition for inclusion criteria or exclusion criteria?

4.     Table 1-2 should be specific to summarize the dose distribution for breast cancer radiotherapy techniques. In Table 1 and Table 2, do all research studies include left side breast cancer only or both sides breast cancer? What’s the meaning of abbreviation of Dmax/FLCA?

5.     In Table 1 and Table 2, what the number of patients (not total patients only) received the different treatment technique being treated? Furthermore, what does meaning the p-value in these Tables?

6.     There is no result section in main manuscript, however, there is a “result” section in the abstract.

7.     In the section of “Discussion”, is there and search strategy to include the topic about relevant sections A) Natural antioxidants, B) Pharmacological drugs, C) Prone positioning…? Please focus on strategies for the prevention of radiation-induced cardiotoxicity in left-sided breast cancer patients, either radiotherapy technique or pharmacological drugs, or specific modality.

8.     The misspelled “myocardial infraction” should be revised as “myocardial infarction.”

9.     In line 311, the word “SBRT” should be revise as “SGRT

10.  The manuscript should be carefully revised in regard to the qualities of the table and the whole manuscript need to be improved.

Reviewer 3 Report

Dear Authors,

In my opinion, the introduction is too broad. There seems to be unnecessary text on well-known BC risk factors. The described radiotherapy methods could be supplemented with illustrations.

Round 2

Reviewer 1 Report

Thank you for having addressed all my requests. Your paper improved a lot and, in my opinion, is now suitable for publication.

Reviewer 2 Report

From a general standpoint, the current review does not offer substantial insight relevant to cardioprotective strategies for the prevention of radiation-induced cardiotoxicity in left-sided breast cancer. Although the revised manuscript is improved, many problems remain. Regarding radiotherapy technique, the current study does not analyze or offer additional mechanistic insight relevant cardioprotective role, and only list many related reported journals in Table 1-2 without any analysis or comparison.

Specific concerns:

1.     Define each abbreviation the first time it is used in the abstract and the first time it is used in the text (i.e., the Introduction through Discussion). i.e., line 48 in Abstract, (organs at risk (OARs)?). After an abbreviation has been defined in the text, do not spell out the word again in the text. i.e., line 62: The majority of breast cancers ?, Do not use an abbreviation (i.e., spell out the word) unless it is used at least three times in the text.

2.     The text should be checked again for grammatical mistakes.
